

# A portable reflected-sunlight spectrometer for $CO_2$ and $CH_4$

Benedikt A. Löw[1], Ralph Kleinschek[1], Vincent Enders[1], Stanley P. Sander[4], Thomas J. Pongetti[4], Tobias D. Schmitt[1], Frank Hase[5], Julian Kostinek[6], and André Butz[1,2,3]

[1]Institute of Environmental Physics (IUP), Heidelberg University, Germany
[2]Heidelberg Center for the Environment (HCE), Heidelberg University, Germany
[3]Interdisciplinary Center for Scientific Computing (IWR), Heidelberg University, Germany
[4]Jet Propulsion Laboratory, California Institute of Technology, Pasadena, USA
[5]Karlsruhe Institute of Technology (KIT), Institute of Meteorology and Climate Research (IMK-ASF), Karlsruhe, Germany
[6]School of Engineering and Applied Sciences, Harvard University, Cambridge, MA, USA

**Correspondence:** Benedikt Löw (benedikt.loew@iup.uni-heidelberg.de)

**Abstract.** Mapping the greenhouse gases carbon dioxide ($CO_2$) and methane ($CH_4$) above source regions such as urban areas can deliver insights into the distribution and dynamics of the local emission patterns. Here, we present the prototype development and an initial performance evaluation of a portable spectrometer that allows for measuring $CO_2$ and $CH_4$ concentrations integrated along a long ($> 10\,\mathrm{km}$) horizontal path component through the atmospheric boundary layer above a target region. To this end, the spectrometer is positioned at an elevated site from which it points downward at reflection targets in the region collecting the reflected sunlight at shallow viewing angles. The path-integrated $CO_2$ and $CH_4$ concentrations are inferred from the absorption fingerprint in the shortwave-infrared (SWIR) spectral range. While mimicking the concept of the stationary CLARS-FTS (California Laboratory for Atmospheric Remote Sensing - Fourier Transform Spectrometer) at Los Angeles, our portable setup requires minimal infrastructures and is straightforward to duplicate and to operate at various places.

For performance evaluation, we deployed the instrument, termed EM27/SCA, side-by-side with the CLARS-FTS at Mt. Wilson observatory (1670 m a.s.l.) above Los Angeles for a month-long period in Apr./May 2022. We determined the relative precision of the retrieved slant column densities (SCDs) for urban reflection targets to 0.36–0.55% for $O_2$, $CO_2$ and $CH_4$, where $O_2$ is relevant for lightpath estimation. For the partial vertical columns (VCDs) below instrument level, which is the quantity carrying the emission information, the propagated precision errors amount to 0.75–2% for the three gases depending on the distance to the reflection target and solar zenith angle. The comparison to simultaneous CLARS-FTS measurements shows good consistency, but the observed diurnal patterns highlight the need for taking into account light scattering to enable detection of emission patterns.

## 1 Introduction

Cities are hotspots for emissions of the greenhouse gases carbon dioxide ($CO_2$) and methane ($CH_4$) (e.g. Edenhofer et al., 2014; Gurney et al., 2021; Mitchell et al., 2022; de Foy et al., 2023). The major sources are traffic and heating related emissions, fossil fuel based electricity production, natural gas leakages, and waste treatment (e.g. Gurney et al., 2019; Sargent et al., 2021; Maasakkers et al., 2022; Huo et al., 2022). At the same time, cities have become a driving force behind climate change



mitigation plans and they have committed to ambitious reductions of their greenhouse gas emissions (e.g. Hsu et al., 2019; Wei et al., 2021; Mueller et al., 2021).

To support the evaluation and verification of these emission reduction plans, urban greenhouse gas measurement networks emerge (e.g. Duren and Miller, 2012; McKain et al., 2015; Turnbull et al., 2019; Turner et al., 2020; Dietrich et al., 2021), measurement campaigns (e.g. Hase et al., 2015; Pitt et al., 2022) aim at demonstrating techniques (e.g. Christen, 2014) for emission monitoring, and satellite sensors have been shown to be able to detect $CO_2$ domes (e.g. Reuter et al., 2019; Ye et al., 2020; Kiel et al., 2021) and localized $CH_4$ leakages (e.g. Maasakkers et al., 2022; de Foy et al., 2023). While the remote sensing

tools operated from satellites or ground typically report the column-averaged dry-air mole fractions of $CO_2$ ($XCO_2$) and $CH_4$ ($XCH_4$), in-situ measurements deliver the local concentrations. The former suffer from limited sensitivity to signals confined to the lowermost atmospheric layers and thus require superb precision to detect minute variations. The latter are prone to be affected by local variations implying limited representativeness on the scale of neighborhoods and city districts. Techniques that average the greenhouse gas concentrations along horizontal paths above cities have been suggested as a tool to bridge

the sensitivity gap between in-situ and column-averaged data (e.g. Fu et al., 2014; Rieker et al., 2014; Queißer et al., 2016; Griffith et al., 2018). The horizontally integrated concentrations promise better sensitivity to surface emissions than column-averaged data. Concurrently, they are more representative on kilometer-scales than urban in-situ data since they average over local emission and turbulence patterns.

The California Laboratory for Atmospheric Remote Sensing (CLARS) - Fourier Transform Spectrometer (FTS), stationed

on Mt. Wilson in the North of the Los Angeles (LA) basin at roughly 1700 m elevation, is an observing system that delivers $CO_2$ and $CH_4$ concentrations integrated along horizontal pathlengths of a few ten kilometers (Fu et al., 2014). The CLARS-FTS points into the LA basin collecting sunlight reflected off the ground from various locations (termed *reflected-sun* configuration in the following). After reflection, the light propagates at a shallow angle quasi-horizontally through the urban boundary layer towards the observer at Mt. Wilson. From each of the collected absorption spectra, the spectral analysis provides path-integrated

$CO_2$ and $CH_4$ concentrations. Pointing sequentially at various reflection locations throughout the LA basin enables horizontal mapping of the urban dome of $CO_2$ and $CH_4$ with a resolution on the district scale in a few hours. Since the technique uses sunlight, its operation is limited to daytime and fair weather conditions. The spectral analysis methods need to take into account that there is a quasi-vertical lightpath component from the sun to the reflection point on the ground in addition to the quasi-horizontal component above the city, and that light scattering by particles in the atmosphere complicates the radiative

transfer considerations (Zhang et al., 2015; Zeng et al., 2017, 2020b). Despite these challenges, the CLARS-FTS has been delivering various insights into the spatiotemporal distribution of $CO_2$ and $CH_4$ (and carbon monoxide (CO), nitrous oxide ($N_2O$), water vapor isotopologues) above the LA basin and the related greenhouse gas emission patterns on the ground (Wong et al., 2015, 2016; He et al., 2019; Zeng et al., 2020a, 2021; Addington et al., 2021)

Here, we present the prototype deployment of a greenhouse gas spectrometer that mimics the observational configuration

of the CLARS-FTS but, unlike the latter, the proposed instrument is portable. Requiring minimal infrastructure, it can be deployed at any location with a vantage point that is elevated above a localized source region and allows for a clear view into the area. The envisaged use-cases are the replication of the CLARS-FTS experiment in places other than LA and the combined





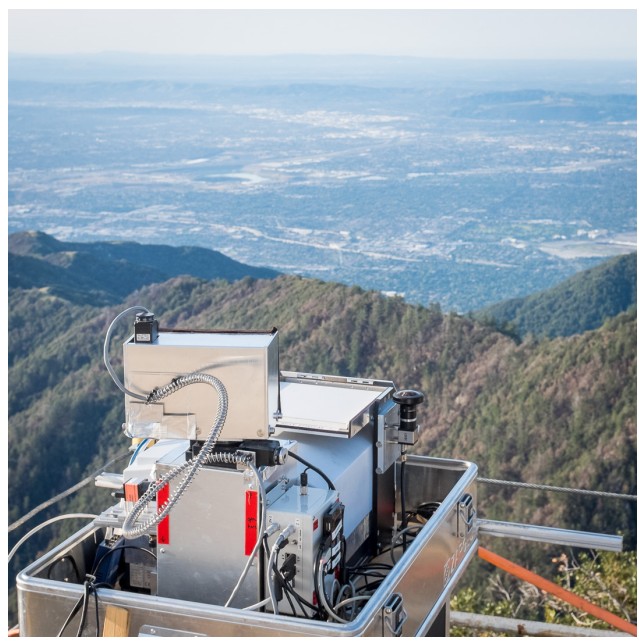

**Figure 1.** The EM27/SCA instrument on top of Mt. Wilson overlooking the Los Angeles Basin. The pointing mirrors are located inside the upper metal protective housing, shielding them from direct sun light. The depicted viewing direction points away from the observer into the basin. The Lambertian reflector plate is mounted on top of the instrument's main body. The context camera is attached on the right.

deployment of several of these instruments, which would allow for crossed-beam configurations to enhance the horizontal mapping capabilities in a primitive tomographic setup. The instrument, called EM27/SCA, is a derivative of the EM27/SUN
FTS, which is in operational use within the COCCON (Collaborative Carbon Column Observing Network) for measuring column-averaged gas concentrations in direct-sun configuration (Frey et al., 2019).

Our study is organized as follows. Section 2 outlines the technical developments and calibration procedures required to make the EM27/SCA work in reflected-sun configuration, while Sect. 3 reports on the conditions of co-deployment with the CLARS-FTS in LA. In Sect. 4, we summarize the spectral retrieval of the target gases $CO_2$, $CH_4$, CO and, for lightpath information, $O_2$
(molecular oxygen). Section 5 discusses the performance of the EM27/SCA and compares it to the simultaneous CLARS-FTS measurements. Section 6 concludes the study with an outlook on future steps.

## 2  Instrumentation - EM27/SCA

The EM27/SUN FTS (Gisi et al., 2012), commercially available from Bruker Optics, forms the basis of our setup. This compact and portable instrument has proven to reliably measure $XCO_2$ and $XCH_4$ in several field campaigns (e.g. Hase et al., 2015;
Butz et al., 2017; Luther et al., 2019, 2022) and is the backbone instrument of the COCCON network (Frey et al., 2019). The FTS has a maximum optical path difference of $\Delta = 1.8\,\mathrm{cm}$, corresponding to a resolution of $1/\Delta = 0.56\,\mathrm{cm}^{-1}$. As the





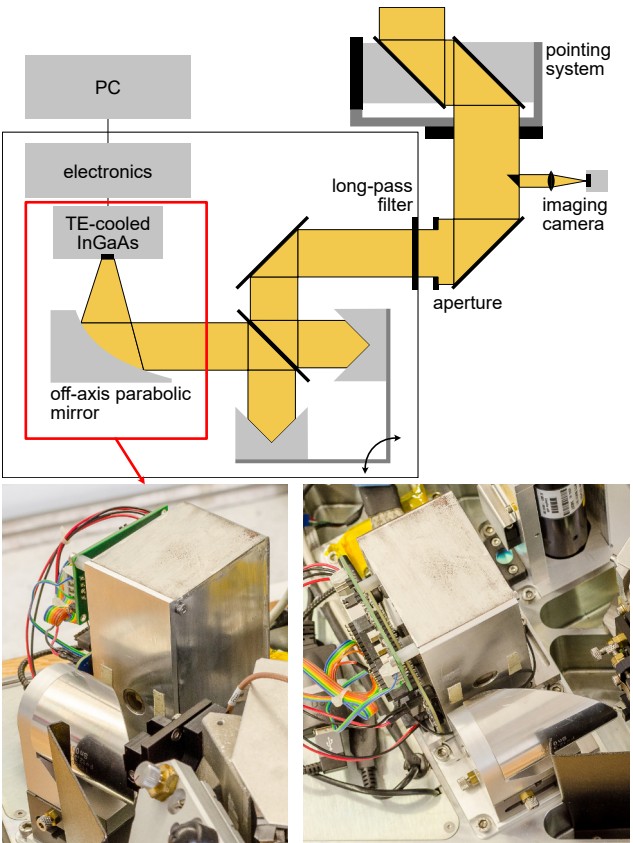

**Figure 2.** Schematics of the lightpath through the EM27/SCA (top panel). Main modifications are the custom detector module (lower panel photographs) and the imaging camera recording the FOV. Adapted from Gisi et al. (2012).

EM27/SUN is designed for direct sunlight observations, we modified several of its components to make it suitable for the fainter radiances expected when measuring sunlight reflected off the ground. Fig. 1 and 2, respectively, show a photograph of the instrument during deployment at LA, and a schematic of the modified spectrometer. We replaced the standard room-
temperature detector module by a 2000 times more sensitive extended InGaAs detector (Hamamatsu G12183-203K) featuring custom-built electronics with a two-stage thermo-electric (TE) cooler set to operating temperatures of -20°C. The spectral cut-off of the detector and a foil filter mounted in front of it limit the sensitive range to 4000–12000 cm$^{-1}$. The 90° off-axis focusing mirror was replaced by one with 44 mm diameter and an effective focal length of $f_{\text{eff}} = 33$ mm. Given a diameter of 0.3 mm of the photosensitive area of the detector diode, the full field-of-view (FOV) is 9.1 mrad (0.52°). We removed all
apertures inside the instrument and mounted an adjustable iris aperture in front of the entrance window. We typically use a beam diameter of 2.6 cm. Together with the fast focusing mirror, this enhances the throughput by a factor of 70. Thus, the more sensitive detector and the increased throughput enhance the sensitivity of the EM27/SCA by a factor $\sim 10^5$ compared to the standard EM27/SUN.





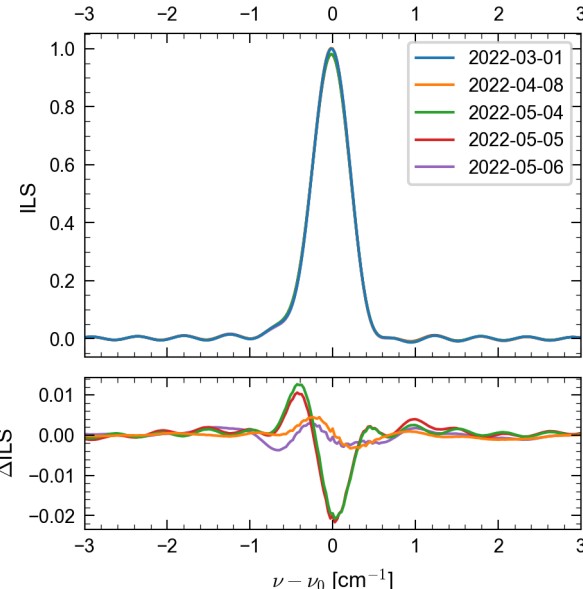

**Figure 3.** ILS measurements before and during the field campaign at Mt. Wilson. The ILS are normalized to unity area and subsequently scaled such that the ILS measured on Mar. 1, 2022 has a peak height of 1. The lower panel shows differences to the ILS measured on Mar. 1, 2022. The differences between measurements are within the expected repeatability.

In front of the spectrometer, the pointing system (derived from EM27/SUN's "sun-tracker") collects sunlight reflected from the ground target into the spectrometer. The pointing system is an alt-azimuthal mount consisting of two rotational stages moving two 50 mm elliptic flat mirrors in azimuthal and zenith direction. Behind the pointing mirrors, a prism couples a small portion of the parallel beam into an imaging camera boresighted with the instrument, allowing for real-time tracking of the instrument's FOV. The FOV-camera is a 1/2" CMOS sensor (1280×1024 px) with a 900 nm longpass filter and an objective lens with a focal length of $f = 100$ mm, representing the instruments FOV with a diameter of 181 px. In addition, an identical context camera with a 185° FOV fisheye lens was mounted on the side of the instrument (see figure 1), recording the general meteorological conditions and skylight context. Finally, we mounted a horizontally levelled Zenith Lite™Lambertian reflector plate on top of the EM27/SCA's main body such that reference observations of reflected sunlight without horizontal path component ("short-cut" observations) can be carried out at any time. The diffuse reflectivity of 50% makes the brightness of the reflector plate comparable to bright ground targets.

## 2.1 Instrument lineshape

We determined the instrument lineshape (ILS) of the EM27/SCA from $H_2O$ absorption lines measured in open-path configuration. Following the procedure described in Frey et al. (2015), we averaged 30 min of spectra of a halogen lamp positioned at a distance of 3.2 m while keeping ambient pressure, temperature and humidity conditions constant. From these spectra, we



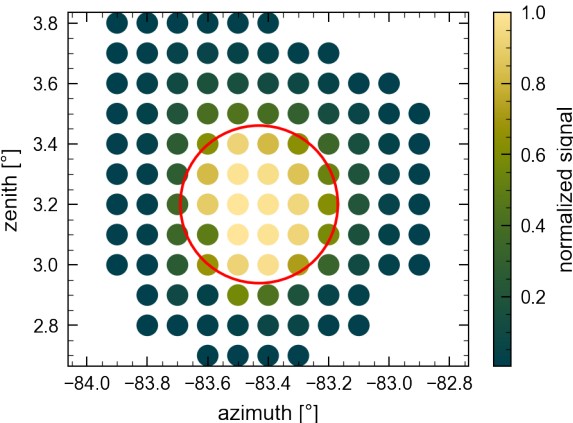

**Figure 4.** Mapping of the FOV of the EM27/SCA by scanning over a point source. The nominal FOV is marked in red around the estimated center.

then retrieved the ILS using the LINEFIT software (Hase et al., 1999) with Norton-Beer medium apodization. The FOV of the
EM27/SCA is considerably larger than the one of the standard EM27/SUN. However, homogeneous illumination of the FOV
is important for an accurate representation of the ILS. We therefore did not point the halogen lamp directly into the instrument,
but instead observed the homogeneously illuminated reflector plate mounted on top of the instrument (see Fig. 1).

Figure 3 illustrates the ILS and its changes monitored during the course of the measurement campaign in LA reported below.
The observed differences are within the expected repeatability of the calibration procedures in the field. The ILS has a full-
width-at-half-maximum of $0.54\,\mathrm{cm}^{-1}$ and is slightly asymmetric. Due to the fast optics and larger FOV, imaging quality of the
EM27/SCA is worse and optical alignment is more delicate than for the standard EM27/SUN. Limiting the beam diameter to
$2.6\,\mathrm{cm}$ as described above clearly improved the width of the ILS and its symmetry.

## 2.2 Field of view and scene heterogeneity

In addition to the FOV of 9.1 mrad ($0.52°$) calculated from the optical parameters, we empirically determined (1) the width of
the FOV and (2) its position within the image of the FOV-camera. To this end, we used a collimated light source with an iris
aperture.

To measure the width of the FOV, we scanned the pointing system in steps of $0.1°$ across the light source positioned at
approximately 18 m distance from the spectrometer. Figure 4 shows the intensity mapping with the nominal FOV overlayed.
The width of the measured FOV is in good agreement with the nominal FOV of $0.52°$ and only slightly elongated along the
zenith direction. Note that the FOV rotates with changes in azimuth viewing direction. To determine the FOV position within
the FOV camera image, we positioned the light source approximately 33 m away from the instrument. We determined the
pointing direction corresponding to the maximum signal. We then scanned in azimuth and zenith direction until the signal





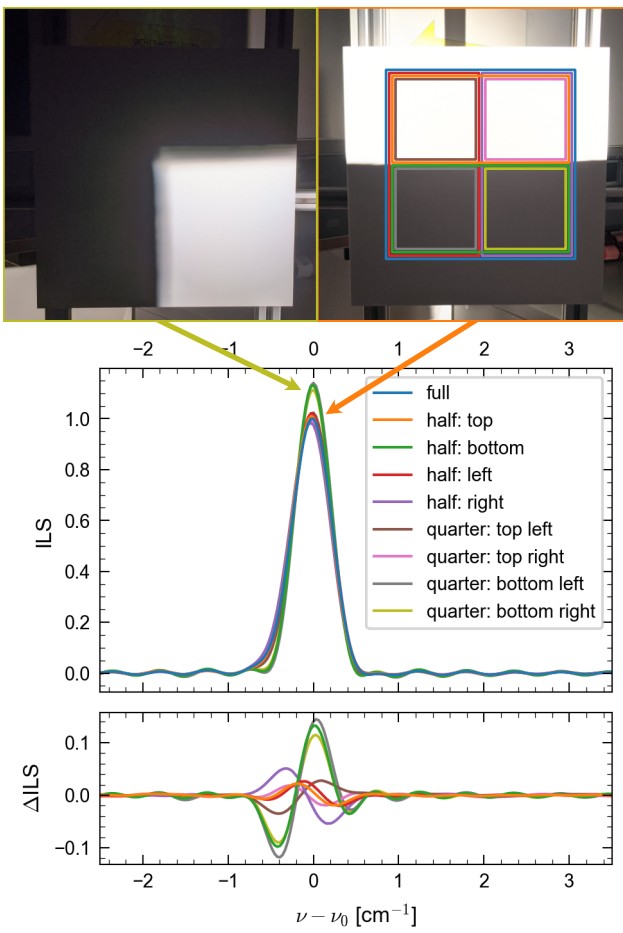

**Figure 5.** Test configuration (upper row) and ILS measurements (bottom) for an inhomogeneous illumination of the reflector plate. The variations of the ILS under different illumination patterns show that the instrument is influenced by scene inhomogeneity in extreme cases.

dropped by half respectively. From this, we determined the center position of the light source in the camera image as the average position of all half-maximum positions and recorded an image pointing there. We determined the center pixel of the

instrument FOV inside this image from fitting an ellipse to the saturated pixels. The off-center position of the camera in the beam together with the finite distance of the light source leads to a parallax error. The camera is placed $2.5\,\mathrm{cm}$ from the beam axis. From that we estimated this error to 10% of the full FOV ($0.05°$). Additionally, we observed that the FOV drifts over the course of a day. The drift is limited to below $0.1°$, and is most likely caused by thermal expansion of the pointing system.

Real world scattering targets are only approximately homogeneous. To determine the influence of scene heterogeneity on

the ILS, we set up a laboratory experiment using a $50{\times}50\,\mathrm{cm}^2$ Lambertian reflector plate located at a distance of $7\,\mathrm{m}$ from the EMS27/SCA. A halogen lamp illuminated the reflector plate homogeneously. We conducted various experiments by shading parts of the reflector plate such that only half or a quarter of the area was illuminated (Fig. 5, upper part). For each configuration,



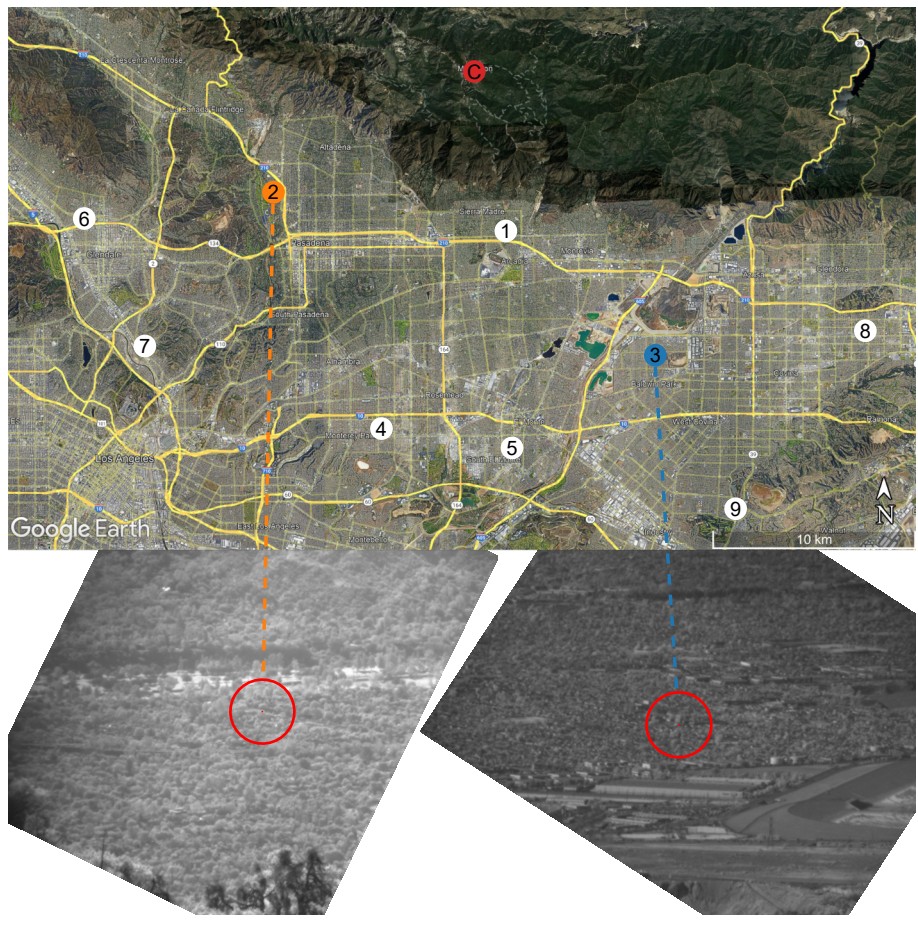

**Figure 6.** Location of the CLARS-FTS on Mt. Wilson (C) and ground scattering targets (numbers) in the LA basin. The lower panel shows FOV-camera images of the West Pasadena (WP) (2, left image) and Baldwin Park (BP) (3, right image) targets. The instrument FOV is marked in red.

we recorded an ILS according to the procedure outlined in Sect. 2.1. Fig. 5, lower part, shows the distortions of the ILS caused by inhomogeneous illumination. Clearly, we observe changes in the width and the peakedness of the ILS with narrower and more peaked ILS occurring when the lower part of the reflector plate was blocked. Note that the orientation of the image depends on viewing direction, since the image rotates with azimuth viewing direction.

We conclude that the ILS of the EM27/SCA is sensitive to scene heterogeneity. However, the checkerboard pattern used for the laboratory testing is certainly an extreme configuration. Thus, we expect the effect to be smaller for typical targets in the field. Nevertheless, we took care of selecting homogeneous targets for our field deployment in LA.



## 3 Field deployment at Los Angeles

To evaluate the performance of the EM27/SCA, we deployed it side-by-side with the CLARS-FTS during a period of roughly one month. We positioned the EM27/SCA adjacent to the CLARS-FTS on a small viewing terrace at the Mount Wilson observatory at 1670 m a.s.l. overlooking the Los Angeles basin (Fig. 1). We took measurements during 26 days in the period from Apr. 7, 2022, to May 5, 2022, pointing at up to 9 targets in the northern LA basin with slant distances of up to 25 km (see map in Fig. 6).

A typical measurement cycle started with a short-cut measurement of the reflector plate and then, cycled through the ground targets in the basin, including one additional short-cut measurement. We co-added 10 interferograms (at an interferogram sampling frequency of 10 kHz) per target resulting in roughly 12 min duration for a full cycle of 9 ground targets plus 2 reflector measurements. In the later part of the campaign, we recorded 3 spectra per target, with 10 interferograms each, increasing the duration of a full cycle to 36 min. The early phase of the campaign was dedicated to testing the setup and finding the best measurement configurations and therefore, on a few days, a cycle covered only a single or a few ground targets (instead of the total of nine) and one reflector measurement. On three days, we only conducted reflector measurements since a marine cloud layer covered the basin. Simultaneously with the spectrometer recording interferograms, the FOV-camera captured an image of each target scene in each cycle and the context-camera recorded an image of the overhead sky.

The pointing directions for each target were determined at the start of the campaign via the FOV-camera of the CLARS-FTS. As discussed in Sect. 2.2, special care was taken that the target scenes were homogeneous i.e. not exhibiting strongly contrasted features across the FOV. However, in a urban setting, perfectly homogeneous targets are not available. GPS coordinates of the targets are available through the pointing calibration of the CLARS-FTS (Fu et al., 2014). After target selection, the EM27/SCA was pointed to the same targets as CLARS-FTS via direct comparison of the FOV images.

For our initial demonstrator assessment in Sect. 5, we focus on the two targets West Pasadena (WP) and Baldwin Park (BP), which have the most revisits. These targets represent urban ground with houses, as visible in Fig. 6. The targets have a slant distance of 11.5 km (WP) and 16.4 km (BP).

## 4 Spectral retrieval

The EM27/SCA recorded DC-coupled interferograms which we converted to absorption spectra via Fourier transform using the preprocessor of the PROFFIT retrieval software routinely employed for EM27/SUN processing within the COCCON network (Frey et al., 2019). Throughout this study, we used the Norton-Beer medium apodization. The processing included a DC correction that corrected the interferograms for mild brightness fluctuations during recording. Together with the spectra, we stored the height of the center burst of the interferograms as well as a measure for DC fluctuations. Later, we used this data to identify spectra suffering from cloud contamination and over-saturation.

Fig. 7 shows a typical absorption spectrum measured with the EM27/SCA and highlights the absorption bands used for the retrievals of $CO_2$, $CH_4$, CO and $O_2$. We submit the absorption spectra to a variant of the RemoTeC radiative transfer and retrieval algorithm to infer the gas abundances. RemoTeC has been designed for satellite measurements of $XCO_2$, $XCH_4$,







**Figure 7.** Typical reflected-sun spectrum of the BP target around 15:50 LT (at SZA 46.6°). The first panel shows the full spectrum with the retrieval windows highlighted in color. A reflector spectrum from roughly the same time is plotted in the background. The other panels show the retrieval windows for $CO_2$ (red), $CH_4$ (orange), CO (purple) and $O_2$ (green). Note that CO is a weak absorber and that W6 and W7 are dominated by $CH_4$ and $H_2O$ absorption lines.

and XCO in the SWIR spectral range (Butz et al., 2011). The variant we employed here has been modified to accommodate the particular observation geometry (see Fig. 8) by dividing the layered, horizontally homogeneous model atmosphere into
an overhead part, i.e. the part of the atmosphere above the instrument stationed at Mt. Wilson, and a boundary layer part, i.e. the part of the atmosphere below the instrument where the targets in the LA basin are located. For the overhead part, the forward model calculates the one-way downward slant transmittance of sunlight at instrument level. For the boundary layer part, the forward model corresponds to the two-way transmittance between instrument level and the level of ground reflection taking into account the solar (SZA, $\theta_s$) and the viewing zenith angles (VZA, $\theta_v$). Combining the overhead and boundary



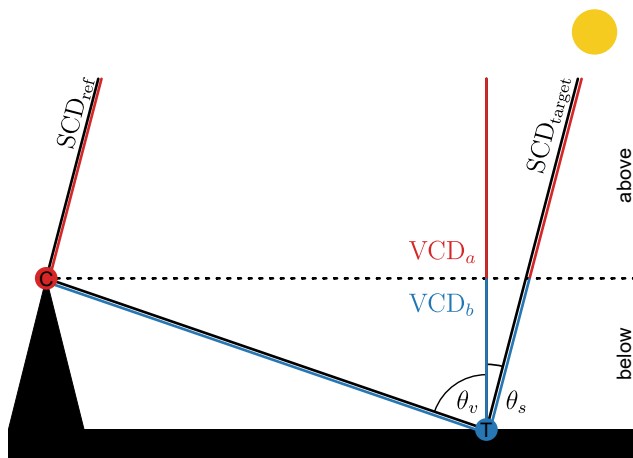

**Figure 8.** Representation of the viewing geometry as implemented in the retrieval. We separate the model atmosphere at instrument level, scaling the VCDs above in reflector and below in target retrievals. From the resulting VCDs we derive the total SCDs for reflector ($\mathrm{SCD_{ref}}$) and target ($\mathrm{SCD_{target}}$) measurement. Finally, we calculate $\mathrm{VCD}_a$ from $\mathrm{SCD_{ref}}$ and $\mathrm{VCD}_b$ from the difference between $\mathrm{SCD_{ref}}$ and $\mathrm{SCD_{target}}$ (see Eq. (3)).

layer transmittances yields the forward model simulating the expected measurements for the LA basin targets. For short-cut measurements of the reflector plate, only the overhead part is relevant.

We set up the retrieval such that a least-squares estimator delivered vertical column densities (VCDs) of the target absorbers. We regularized our retrieval such that each absorber had one degree of freedom in the vertical. For reflector measurements, we scaled the overhead VCDs. For target measurements, we varied the partial VCDs in the boundary layer part, while the

overhead part was imposed by the a priori. The following performance analysis makes further use of slant column densities (SCDs) defined as

$$\mathrm{SCD} = \mathrm{VCD} \times \mathrm{AMF} \tag{1}$$

where AMF is the air mass factor, the ratio of the slant path through a layer to the vertical depth of the layer. As such, the AMF is different for the parts above ($\mathrm{AMF}_a$) and below ($\mathrm{AMF}_b$) instrument level. They are defined via

$$\mathrm{AMF}_a = \frac{1}{\cos(\theta_s)}, \ \ \mathrm{AMF}_b = \frac{1}{\cos(\theta_s)} + \frac{1}{\cos(\theta_v)} \tag{2}$$

under our geometric assumptions (c.f. Fig. 8). Analyzing the SCDs is useful, because they directly relate to optical thickness, and with it to the spectra, making them a good quantity for the performance evaluation of the EM27/SCA.

VCDs offer the advantage that geophysical variation is deconvolved from changes in the geometric lightpath. However, they incorporate geometric assumptions, which in our case are not uniform throughout the model atmosphere, making total

VCDs difficult to interpret. To circumvent this problem, we evaluated the partial VCDs above ($\mathrm{VCD}_a$) and below ($\mathrm{VCD}_b$) the instrument separately. $\mathrm{VCD}_a$ are directly calculated from the reflector measurements. $\mathrm{VCD}_b$ are derived from the differences



**Table 1.** Retrieval windows

| ID | spectral range [cm$^{-1}$] | width [cm$^{-1}$] | absorber target | interfering |
|----|----|----|----|----|
| W1 | 7765 – 8005 | 240 | $O_2$ | $H_2O$, $O_2$ CIA |
| W2 | 6180 – 6260 | 80 | $CO_2$ | $H_2O$ |
| W3 | 6297 – 6382 | 85 | $CO_2$ | $H_2O$ |
| W4 | 5880 – 5996 | 116 | $CH_4$ | $H_2O$ |
| W5 | 6007 – 6145 | 138 | $CH_4$ | $H_2O$, $CO_2$ |
| W6 | 4208.7 – 4257.3 | 48.6 | CO | $H_2O$, HDO, $CH_4$ |
| W7 | 4262.0 – 4318.8 | 56.8 | CO | $H_2O$, HDO, $CH_4$ |

between the SCDs from the target ($SCD_{target}$) and the reflector ($SCD_{ref}$), taking into account $AMF_b$.

$$VCD_a = \frac{SCD_{ref}}{AMF_a}, \quad VCD_b = \frac{SCD_{target} - SCD_{ref}}{AMF_b} \tag{3}$$

To calculate $VCD_b$, we interpolated $SCD_{ref}$ between reflector measurements bracketing the time of the target measurement.
$VCD_b$ is the quantity ultimately of interest and also the quantity our target measurements are most sensitive to.

Note that, for the initial assessment of the instrument performance, we did not consider scattering by particles in the forward model, as also previously assumed by Fu et al. (2014).

Table 1 lists the spectral windows (see also Fig. 7) from which we retrieved the $O_2$, $CO_2$, $CH_4$ and CO VCDs. Absorption cross sections for all species were calculated from spectroscopic parameters using a Voigt line shape and the HITRAN 2020
database (Gordon et al., 2022). All absorbers were retrieved independently for each window. Note that windows W4 and W5 excluded the $CH_4$ Q-branch, as we found systematic errors in retrievals including it. For windows W6 and W7, we found that retrieving the main $H_2O$ isotopologue and HDO independently reduced fitting residuals. In all other windows the $H_2O$ cross sections included lines of all isotopologues scaled with their standard abundance. Broad band variation of the spectrum baseline was accounted for through a polynomial of 4th order in W2 to W7 and 5th order in the largest window W1. We
computed the solar spectrum from the empirical line list by Toon (2015), which was also used by Coddington et al. (2021) for the computation of the TSIS-1 HSRS.

RemoTeC used a priori profiles from CarbonTracker 2019 (Jacobson et al., 2020) for $CO_2$ and TM4 (Meirink et al., 2006) for $CH_4$ and CO. Water vapor ($H_2O$) and oxygen ($O_2$) a priori profiles were calculated from NCEP meteorological data (NCEP, 2000), which also provided the vertical pressure and temperature profiles. The pressure at instrument level was taken from the
CLARS weather station (Fu et al., 2014) and was used as the lowest pressure level for the reflector measurements as well as for dividing the atmosphere in the overhead and boundary layer part in the case of the target measurements.

To exclude unreliable measurements, we applied a series of filters to all measurements. We did not analyze spectra recorded at SZA greater than 70°, and filtered measurements with signal-to-noise ratio (SNR, defined in Sect. 5) less than 100 in W1–





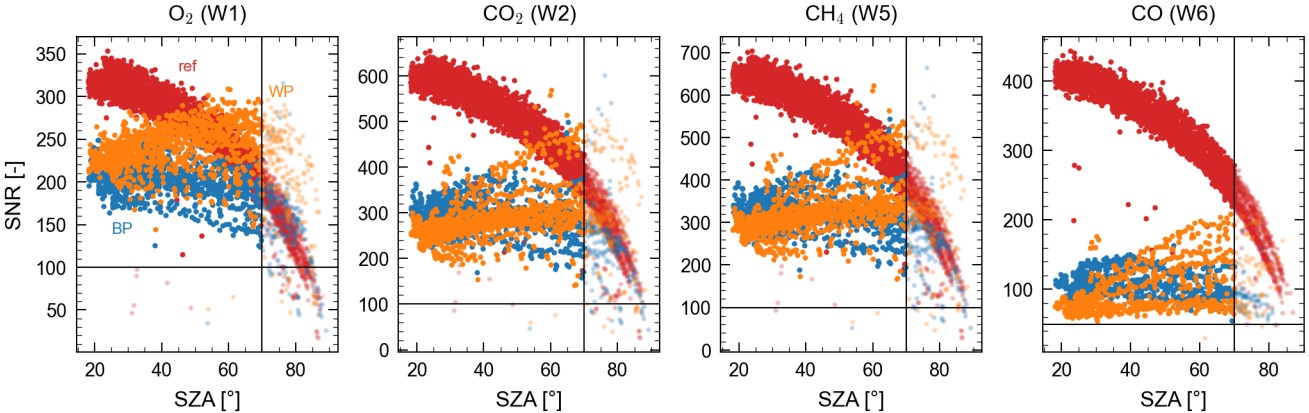

**Figure 9.** SNR for spectra measured between Apr. 7 and May 5, 2022 plotted against SZA for the reflector (red), and the targets BP (blue) and WP (orange).

W5 and less than 50 in the CO retrieval windows. Only few measurements were removed due to low SNR in W1–W5 (0.2%), while 4% of the measurements were excluded from CO retrievals in W6 and W7. Additionally, the CO retrieval did not fully converge for 18% of the measurements. These measurements were still used for retrievals of species from the other windows. We further filtered for cloud contamination by utilizing the DC component of the interferogram, as proposed by Klappenbach et al. (2015). Since the DC component changes with temporal variations in sunlight intensity, its variability increases when the sun is partially obstructed. We calculated the variability $DC_{var}$ as the fraction of maximum ($DC_{max}$) to minimum ($DC_{min}$) value of the baseline and subtracted one so that no variability corresponds to a value of zero.

$$DC_{var} = \frac{DC_{max}}{DC_{min}} - 1 \tag{4}$$

The cut-off value of this filter was empirically determined to 0.03 from cloud free measurements, identified with the help of the context camera. 7% of the measurements were removed by this filter.

## 5 Performance

### 5.1 Signal-to-noise and spectral residuals

To evaluate the performance of the EM27/SCA, we start with calculating the spectral SNR for the retrieval windows defined in table 1 and illustrated in Fig. 7. The SNR is the ratio of the maximum transmittance in the respective window over the out-of-band noise. The latter was calculated as the standard deviation in the spectral range 2500–3056 $cm^{-1}$, where there is zero throughput of the spectrometer system. Generally, the SNR depends on multiple factors including the spectrally variable optical throughput, sensitivity of the detector, reflectivity of the target, spectral solar irradiance, position of the sun in the sky,





and atmospheric conditions. Fig. 9 shows the SNR in dependence of SZA for spectra of the reflector and the two targets BP and WP.

Typical SNR values range from 200–400 for target and 400–600 for reflector measurements. The SNR for the reflector measurements is most compact and follows a clear dependence on SZA since these measurements, being conducted at high elevation and having zero horizontal path, approximate direct sun conditions. The SNR of the target measurements shows more scatter and no compact SZA dependency, which points at a strong influence of atmospheric conditions, atmospheric scattering effects and the geometry of surface structures. For the $O_2$ window (W1), the generally greater surface reflectivity compensates for the reduced detector sensitivity compared to the $CO_2$ (W2, W3) and $CH_4$ windows (W4, W5) of the targets. The CO retrieval windows (W6, W7) exhibit lower SNR on the order of 50–150, as the windows are located towards the longwave cut-off of the InGaAs detector. Note that SNR in the CO retrieval windows is generally lower for WP than for BP. We attribute the upward SNR trends with increasing SZA for WP to atmospheric scattering towards the evening.

Processing the acquired spectra with RemoTeC (see Sect. 4) yields best-fit simulated spectra. They allow for inspecting the spectral fitting residuals, defined as the differences between the measured spectra and the best-fit simulations. Fig. 10 shows the spectral residuals for one window of each species, averaged over a whole day of observations. For averaging, each residual was normalized by the maximum measurement signal in the respective window. Due to the averaging, noise is negligible and the spectral residuals reveal systematic errors due to deficiencies in the retrieval simulations, such as forward model approximations and parameter errors. The residuals and their root-mean-square (RMS) for the reflector measurements are generally smaller than for the targets in the LA basin. This points at contributions from the neglect of atmospheric scattering, which is a better approximation for the reflector than for the target measurements. Further, spectroscopic uncertainties, such as line shape errors, typically grow with the length of the lightpath, as absorption lines become more opaque and thus target spectra might be more affected than the reflector measurements. Individual spectral structures correlate with the positions of $H_2O$ lines (e.g. at 5892, 5940, 6020, 6226, 6242, 6305, 6325, 6334 and 6363 $cm^{-1}$), suggesting spectroscopic errors. The fitting residuals in the $O_2$ window (W1) are larger and more systematic than for $CO_2$ (W2, W3) and $CH_4$ (W4, W5). Despite revisiting our calibration procedures, taking into account spectral variation of the ILS and preforming various sensitivity runs, we were not able to identify the cause for this.

Fig. 11 evaluates the RMS of the spectral residuals against SNR. Generally, the RMS follows the 1/SNR pattern expected from a noise-dominated measurement for both reflector and target spectra. However, there is an offset that comes from the systematic residual structures which are particular pronounced in the $O_2$ band.

Overall, the spectral performance of the EM27/SCA is promising in terms of retrieving precise $CO_2$, $CH_4$ and $O_2$ columns. For CO, the spectral cut-off of the detector and the low surface reflectivity around 4250 $cm^{-1}$ imply small SNR. A next-generation instrument might consider detectors and optics with better performance at the longwave end of the SWIR.

## 5.2 Precision

To estimate the measurement precision we empirically evaluate the precision of the retrieved SCDs for each gas (as explained in Sect. 4). We evaluated 4 days of measurements from the early campaign period, where the sampling of the evaluation targets





**Figure 10.** Spectra and spectral residuals for the reflector (upper row), target WP (middle row) and targt BP (lower row) for the W1 ($O_2$, green, first column), W2 ($CO_2$, red, second column), W5 ($CH_4$, orange, third column) and W6 (CO, purple, fourth column) windows. The lower subplots in each panel show a single spectrum; the upper subplots show a residual from the single spectrum (colored) and a full day average (black) of 124 (reflector), 124 (WP), and 122 (BP) normalized residuals (119, 112 and 112 for CO).

was densest. We calculated the precision error as the standard deviation $\sigma$ of the ratio between each measurement and a rolling average,

$$\Delta\mathrm{SCD} = \sigma\left(\frac{\mathrm{SCD}_i}{\langle\mathrm{SCD}\rangle_t}\right) \tag{5}$$

where $\Delta\mathrm{SCD}$ is the estimated relative precision error of the SCDs, $\mathrm{SCD}_i$ is the slant column density of measurement $i$, and $\langle\mathrm{SCD}\rangle_t$ is the rolling average SCD. We chose an averaging interval of $t = 30$ min, corresponding to 7 measurement cycles, as

a trade-off between convolving actual geophysical variability into our precision estimates and considering a sufficient number



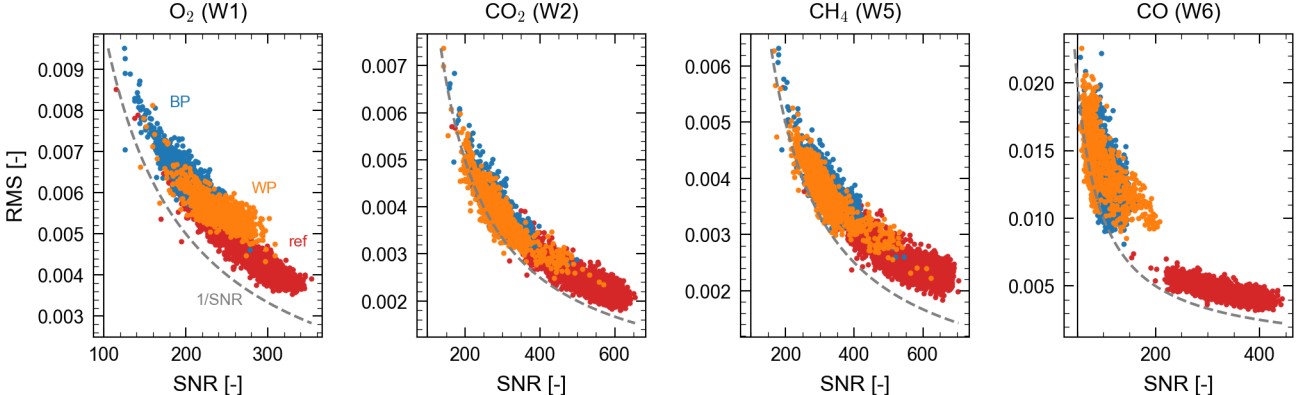

**Figure 11.** RMS for spectra measured between Apr. 7 and May 5, 2022 plotted against SNR for the reflector (red) and the BP (blue) and WP (orange) targets.

**Table 2.** Relative precision error of the absorber SCDs estimated from the departures of individual measurements from a 30 min rolling average. Numbers in parentheses show the last digit changes when repeating the analysis with 20 min and 40 min rolling averages.

|  | window | reflector | WP | BP |
|---|---|---|---|---|
|  |  | N = 334 | N = 409 | N = 363 |
| $O_2$ | W1 | 0.49(1)% | 0.40(3)% | 0.51(5)% |
| $CO_2$ | W2, W3 | 0.36(2)% | 0.41(3)% | 0.39(3)% |
| $CH_4$ | W4, W5 | 0.50(2)% | 0.55(4)% | 0.49(5)% |
| CO* | W6, W7 | 7.5(4)% | 15.6(9)% | 8.7(5)% |

*N is lower for CO retrievals: 261 (reflector), 177 (WP), 232 (BP)

of samples for calculating the average SCDs. Fig. 12 shows the ratio of the SCDs to their rolling average against SZA, as well as the histograms. Table 2 lists the resulting relative precision errors for the various absorbers. For $CO_2$, $CH_4$ and $O_2$ SCDs we find a relative precision of 0.36–0.55% without a distinct dependency on SZA. The precision for CO SCDs is considerably worse, as expected from the low SNR in windows W6 and W7 and the weakness of the CO absorption signal. To test the robustness of these estimates, we repeated the analysis with rolling averages over 20 min (5 cycles) and 40 min (9 cycles), which resulted in only small changes (table 2).

## 5.3 Comparison to CLARS-FTS

To assess the consistency of the EM27/SCA measurements to those of CLARS-FTS, we submitted the CLARS-FTS spectra of the same targets WP and BP and CLARS-FTS' own reflector to the RemoTeC retrieval with analogue settings as for the EM27/SCA spectra. We interpolated the higher precision CLARS-FTS partial VCDs linearly in time to EM27/SCA measure-





**Figure 12.** Departures of $O_2$, $CO_2$, $CH_4$ and CO (columns left to right) SCDs from their 30 min rolling average, plotted against SZA (left subpanels) and as histograms (right subpanels) including a normal distribution with the corresponding mean and standard deviation for the reflector (upper row) and the WP (middle row) and BP (lower row) targets. The scatterplots are color coded by day.

ment instances, and subsequently calculated the CLARS-FTS SCDs. Evaluating all the reflector measurements in the campaign period, we found that CLARS-FTS SCDs differ from EM27/SCA SCDs on average by a factor 1.046 for $O_2$, 1.0068 for $CO_2$, 1.014 for $CH_4$ and 0.997 for CO. Such scaling factors might originate from the different spectral resolutions of the EM27/SCA $(0.56\,\mathrm{cm^{-1}})$ and the CLARS-FTS $(0.2\,\mathrm{cm^{-1}})$ as Gisi et al. (2012) found for TCCON to EM27/SUN comparisons. For all the following analyses, we scaled the EM27/SCA retrievals by these factors.

Fig. 13 shows the correlation between the EM27/SCA and CLARS-FTS retrievals. Measurements of the reflector and both targets show a good correlation in $O_2$, $CO_2$ and $CH_4$ with correlation coefficients near 1. CO shows a decent correlation for reflector measurements. For target CO measurements the low precision dominates.



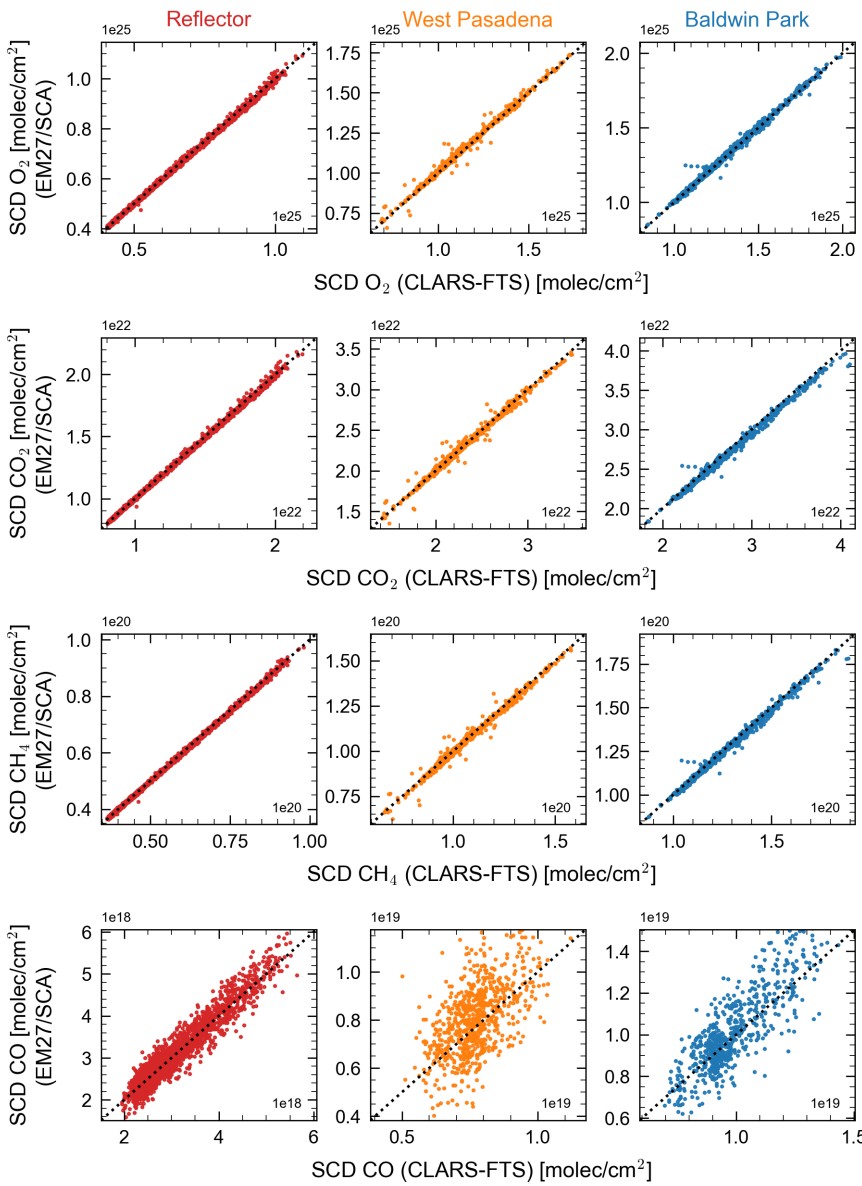

**Figure 13.** Correlation of $O_2$, $CO_2$, $CH_4$ and CO SCDs (top to bottom row) between EM27/SCA and CLARS-FTS for reflector (red, left), WP (orange, middle) and BP (blue, right) measurements. The dotted lines show the 1:1 relationship.

Fig. 14 shows the observed diurnal variations of the $O_2$, $CO_2$, $CH_4$ and CO partial VCDs above ($VCD_a$) and below ($VCD_b$)

instrument level for the EM27/SCA and CLARS-FTS for three illustrative days. We calculate the precision error of the partial VCDs via Gaussian error propagation from the relative precision of target and reflector SCDs (table 3). Since the fraction of



**Figure 14.** Timeseries of the $VCD_a$ (above) and $VCD_b$ (below instrument level) of $O_2$, $CO_2$, $CH_4$ and $CO$ (top to bottom) measured by the EM27/SCA (dots) and the CLARS-FTS (triangles). Three illustrative days are shown left to right. The partial $VCD_a$ is retrieved from reflector measurements (red). The $VCD_b$ below is derived from the difference of WP (orange) and BP (blue) to the reflector measurement. The EM27/SCA precision error is displayed as shading around the rolling average. The first two days belong to the period where we cycled through three targets only, and therefore, these days show a higher sampling rate.





**Table 3.** Relative precision error range for $VCD_a$ and $VCD_b$ on Apr. 13.

|  | $VCD_a$ | $VCD_b$ | |
| --- | --- | --- | --- |
|  |  | WP | BP |
| $O_2$ | 0.49% | 0.77–1.6% | 0.85–1.3% |
| $CO_2$ | 0.36% | 0.75–1.5% | 0.64–0.99% |
| $CH_4$ | 0.50% | 0.98-2.0% | 0.79–1.2% |
| CO | 7.5% | 20–44% | 11–18% |

the light path below instrument level depends on target distance and SZA, the precision error for target measurements increases for closer targets and higher SZA.

Focusing on the overhead reflector measurements first (red symbols in Fig. 14), we find that the two instruments generally
agree well for all four absorbers and that the $VCD_a$ are not much variable during the day and from day-to-day as expected for the quasi-direct sun geometry. Generally, the CLARS-FTS has substantially better precision than the EM27/SCA as expected from the CLARS-FTS evaluation by Fu et al. (2014). The standard deviations of the differences between EM27/SCA and CLARS-FTS $VCD_a$ amount to 1% for $O_2$, $CO_2$ and $CH_4$ respectively, and 7% for CO. There are some systematic deviations, especially in the early morning hours, for $CO_2$ and, although weaker, also for $O_2$ and $CH_4$. For these species, the EM27/SCA
records show a low bias on Apr. 13, and on May 5 there is a diurnal variation on the order of the precision error, which is not present in the CLARS-FTS data. For CO, the large scatter masks possible systematic differences.

Turning the focus to the $VCD_b$ in the boundary layer below instrument level (orange and blue symbols in Fig. 14), we generally find that $O_2$, $CO_2$ and $CH_4$ show strong diurnal variations with amplitudes exceeding 10% of the $VCD_b$. These variations are present in $O_2$ and strongly correlated between species, thus pointing to the influence of light scattering by
aerosols. Since aerosol scattering effects are unaccounted for in our retrieval, the presence of aerosols implies a forward model error that shows up as VCD underestimation if scattering induces lightpath shortening and vice versa for lightpath enhancement compared to our geometric lightpath assumption. While these atmospheric scattering effects have large implications for how to interpret the data (Zhang et al., 2015), the $VCD_b$ from EM27/SCA and the CLARS-FTS are largely consistent. For the BP target (blue symbols in Fig. 14), there are systematic deviations of the EM27/SCA records from the CLARS-FTS data e.g.
on Apr. 13 for $O_2$ in the evening and for $CO_2$ in the morning, and on Apr. 14 for $CO_2$ throughout the day. These systematic deviations are less pronounced for the WP target. After extensive sensitivity studies, we speculate that scene heterogeneity for the BP target may contribute to these deviations. Scene heterogeneity can be variable during the day due to the changing illumination conditions of the ground scene. As explained in Sect. 2.1, heterogeneous illumination of the target affects the ILS of the EM27/SCA, which in turn induces errors in the gas retrievals if the ILS distortion is not accounted for. Due to the lower
spectral resolution of the EM27/SCA, the effect on the retrievals is larger than for the CLARS-FTS since the overall impact of the ILS is larger for low than for high resolution.





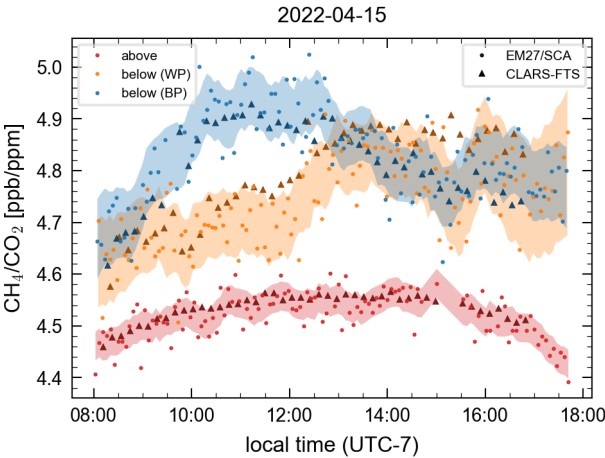

**Figure 15.** Single day of $CH_4/CO_2$ ratio measurements show a $CH_4$ enhancement in the Los Angeles basin assuming a constant $CO_2$ background (Wong et al., 2015).

## 6 Discussion and conclusion

Our study discusses the prototype design and performance of the EM27/SCA FTS. The instrument concept is based on the direct-sun EM27/SUN FTS, but it achieves manifold higher sensitivity making it suitable for measurements of sunlight reflected by ground targets of opportunity. While being portable, the instruments mimics the observation concept of the CLARS-FTS stationed at Mt. Wilson looking downward into the LA basin. Our performance evaluation during a month-long side-by-side deployment with the CLARS-FTS shows that the precision errors for the SCDs of $CO_2$, $CH_4$, and $O_2$ retrieved from EM27/SCA measurements are on the order of 0.5%, which translates into errors of 0.7 to 2% for the partial VCDs in the layer below instrument level ($VCD_b$) in the LA basin, somewhat depending on the ground reflection target. For CO, precision errors for the SCDs are in the range from 7 to 16%, translating into errors of several ten percent for the $VCD_b$ of the targets in the LA basin. Thus, while performance is promising for detecting urban enhancements of $CO_2$ and $CH_4$, the EM27/SCA requires reduction of the noise level in the CO channel (around 2.350 nm).

As previous studies for CLARS-FTS (Zhang et al., 2015; Zeng et al., 2017) and our Fig. 14 show, the particular viewing geometry, with a long horizontal lightpath component exceeding 10 km through the urban boundary layer of LA, causes light scattering by aerosols to be the dominant source of variation in the retrieved $VCD_b$, if aerosol effects are neglected in the gas retrievals. To correct to first order for these scattering effects, the column averaged dry-air mole fractions $X_i$ of gas $i$ can be calculated via

$$X_i = \frac{SCD_i}{SCD_{REF}} \times X_{REF} \tag{6}$$

where REF denotes a reference gas whose dry-air mole fraction $X_{REF}$ is known a priori. The idea is that aerosol scattering effects induce the same relative errors in the spectral retrievals of species $i$ and REF and thus, the effects cancel in Eq. (6).



Using $O_2$ as a reference gas is appealing since its dry-air mole fraction is well known and constant for all relevant purposes here. However, Zhang et al. (2015) show that using $O_2$ as reference gas results in erroneous corrections since the $O_2\Delta$ band is spectrally distant from the retrieval windows of the target gases $CO_2$, $CH_4$ and CO. Thus, spectral variation of the aerosol and ground scattering properties make the radiative transfer in the $O_2$ windows different from the other windows. In contrast,

ratioing $CH_4$ with $CO_2$ from the neighboring spectral windows better compensates for the radiative transfer effects. Fig. 15 illustrates a case where the $CH_4/CO_2$ ratio for BP in the morning has a substantial enhancement compared to the data for WP and the reflector. It appears that the early enhancement for BP arrives later at WP. Following the studies for CLARS-FTS (Wong et al., 2015, 2016), we argue that this is a geophysical signal related to emission and transport patterns in the LA basin. However, in order to relate the $CH_4/CO_2$ patterns to $CH_4$ emissions and their transport, according to Eq. (6), we would

require assumptions on the reference $XCO_2$. The usage of the $CH_4/CO_2$ ratio has been discussed in depth for the CLARS-FTS measurements (Wong et al., 2015, 2016; He et al., 2019) and, we will follow up on such cases in future studies.

Assuming that atmospheric scattering effects can be reliably corrected, the $VCD_b$ errors listed in table 3 and shown in Fig. 14 directly yield a limit for detectable enhancements of the respective gas abundances in the LA basin. Verhulst et al. (2017) found typical local enhancements in the LA boundary layer in the range of 17-31 ppm $CO_2$ and 140-220 ppb $CH_4$. Zhang

et al. (2015) estimated 10–20 ppm $XCO_2$ enhancements for CLARS-FTS measurements of the WP target by calculating the differences between CLARS-FTS spectralon and LA TCCON measurements. Zeng et al. (2021) showed typical enhancements of 20 ppm in $XCO_2$ and 150 ppb in $XCH_4$ retrieved from CLARS-FTS measurements with the GFIT3 full physics algorithm. Assuming background mole fractions of 420 ppm for $CO_2$ and 1900 ppb for $CH_4$, these typical enhancements amount to relative variations of 5% ($CO_2$) and 8% ($CH_4$), and thus, they are greater than our $VCD_b$ errors. If they occur in a correlated

pattern, e.g. due to transport effects such as shown in Fig. 15, the EM27/SCA is able to reliably measure them.

Our analysis suggests further refinements of the instrument. Enhancing signal-to-noise would be required for useful CO measurements, and it would be generally beneficial for more reliable measurements and lower detection limits for all the gases. Section 2.1 highlights that the alignment of the detector optics is cumbersome and that the ILS is somewhat asymmetric, which we mainly relate to the detector optics being optimized for throughput, accepting inferior focusing quality as trade-off.

Further, we find that the ILS of the EM27/SCA is sensitive to inhomogeneous illumination of the FOV (Sect. 2.1) and in Sect. 5, we speculate that residual systematic differences between the EM27/SCA and CLARS-FTS might be related to scene inhomogeneity. Thus, a next generation instrument should consider improved detector optics and homogenizing the FOV.

The largest potential for improvement lies in the explicit integration of aerosol scattering in the retrieval algorithm, as shown by Zeng et al. (2021). This would most likely improve fit quality, reduce uncertainty and pave the way to reliable measurements

of near-ground $CO_2$ and $CH_4$ concentrations.

*Data availability.* The data are available from the corresponding author upon request.



*Author contributions.* BAH developed the instrument configuration and carried out the formal data analysis. BAH, RK, and VE together with TJP conducted the field-deployment at Mt. Wilson. RK, TDS and FH supported the instrument developments. JK built an early prototype of the instrument. TJP and SPS provided CLARS-FTS measurements, supported data analysis and the field-deployment. BAH and AB wrote the manuscript with comments from all co-authors. AB conceptualized the project.

*Competing interests.* Some authors are members of the editorial board of AMT. The peer-review process was guided by an independent editor, and the authors have also no other competing interests to declare.

*Acknowledgements.* The research presented here has been funded by the Deutsche Forschungsgemeinschaft (DFG, German Research Foundation), project number 449857152, and further supported by DFG INST 35/1503-1 FUGG (SDS@hd) and the Ministry of Science, Research and the Arts Baden-Württemberg (MWK).



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
