# Peer review of "A portable reflected-sunlight spectrometer for CO2 and CH4"

_Atmospheric Measurement Techniques, 2023_

## Referee Comment (RC2)

[referee-annotated manuscript omitted]

---

## Author Comment (AC1)

**RC1: 'Comment on amt-2023-101', Anonymous Referee #1, 04 Jul 2023**

https://doi.org/10.5194/amt-2023-101-RC1

**GENERAL COMMENTS**

**The large portions of the GHG gases emit from global mega cities and point sources. Remote sensing form upper sky is a powerful tool to capture potential emission plumes, however, the amount of data with satellites and airplanes is limited. Local time of the existing satellites are around noon. The emission estimates from satellite data have large uncertainties due to local weather conditions such as wind speed and direction. Diurnal variation data from the fixed point will improve emission monitoring. The modification of the light source from a direct solar light (COCCON) to scattered light instrument and its characterization tests are well described in the manuscript. Technical portions are good. However, scientific discussions are needed. I have the following general comments. The discussions on additional characterization and applications will improve the scientific significance. Major revisions are needed.**

We thank the reviewer for the appreciation of our work and the helpful comments. Find our point-by-point reply below.

**(1) Retrieval**

**The present retrieval seems to be modification of the direct solar measurement such as COCCON to slant column densities. Aerosols over LA Basin causes large fluctuations with the large off nadir geometry. Are the authors planning to retrieve several parameters such as aerosol optical thickness, surface albedo, surface pressure from EM27 spectra?**

Yes, we plan to investigate the impact of aerosols on our measurement. For future studies, we will develop a simultaneous retrieval of GHGs and aerosol properties plus surface albedo from the EM27/SCA spectra. However, this is a major complication. Our radiative transfer and retrieval software (RemoTeC) can, in principle, treat aerosol scattering but, for technical reasons (see reply to comment RC2), it cannot treat observers that are positioned inside the atmosphere (as opposed to at top-of-the-atmosphere such as a satellite). The related developments are not within the scope of our instrument performance assessment, here, but they deserve an extra study (and extra time). We added a corresponding paragraph to the discussion section of the manuscript.

**Does the surface reflectance over the LA basin has strong dependency on solar zenith angles? Are there critical angles in viewing and solar zenith?**

We retrieve a spectral background polynomial (c.f. L203). This removes the broadband effects originating from spectral surface reflectance for each measurement individually. If there were critical angles in terms of BDRF angle dependencies, they would show up, for example, in SNR dependencies on SZA or VZA, as the signal level directly relates to surface reflectance. We do not find such cases (c.f. Fig. 9). Additionally, since we neglect aerosol scattering within our retrieval, the directionality of surface reflectance is not incorporated in our forward model. Therefore, the retrieved background polynomial represents the surface brightness for the current viewing geometry. This would be different if the forward model included multiple scattering e.g. between an aerosol layer on the ground.

A remaining possibility for the directionality of surface reflectance to impact our retrieval would be that specular reflection spots apear within the FOV, adding to scene inhomogeneity within the FOV.

As mentioned in the manuscript, we selected the ground scattering targets to exhibit as homogeneous reflectance as possible (c.f. L134). We cannot identify critical angles due to SZA or VZA dependent scene inhomogeneity in the SZA dependence on fit RMS (Fig. AC1). For individual measurements, we occasionally observe point-like specular reflections in the imaging camera images, but their contribution to the overall signal is negligible.

[Figure]

*Figure AC1: RMS over SZA for the different retrieval windows. We do not observe distinct angles critical for the fit quality.*

**Discussions on which parameters are retrieved and assumed will suggest the importance of the measurements.**

As specified in L177f. we retrieve absorber column densities with one degree of freedom simultaneously with a broadband background polynomial (L203). For reflector measurement this is equivalent to a column scaling approach employed by GFIT and PROFFIT (COCCON). For the LA basin target observations however, only the column below instrument level is scaled. This is, because we expect the main variability to occur in this portion of the light path: (1) the emission signal is located there, (2) the pathlength in this region is longest and (3) light path modifications arising from aerosol scattering also arise mainly there. Given this implementation, errors in the a priori above instrument level are mistaken as changes in the lowest layer, which is not a problem since we use only the total SCD for further analysis (c.f. Eq (3)). The remainig error is caused by attributing the molecules in the total column to regions with different pressure and temperature. The alternative implementation of scaling the entire profile would attribute the variability coming largely from below instrument level to the entire total column which would lead to substantially larger errors.

We added this discussion as paragraph after L195 and an overview table summarizing retrieved and assumed parameters to the manuscript.

**(2) Instrument Resources**

**The EM25 spectrometer is still heavy and expensive, if we install several systems from different location. Is it possible to reduce size and weight by relaxing spectral resolution?**

We consider the EM27 spectrometer quite portable and versatile, given that one instrument alone can remotely sample various locations for mapping entire regions.

Resolution is not easily tradable for size and weight, as the maximum optical path difference (OPD), on which the FTS resolution mainly depends, has only a minor contribution to the overall instrument size and weight as long as OPDs are on the order of a few cm (as typical for our case). Reducing the

maximum OPD further from 1.8 cm will have a neglible influence on instrument size and weight. There is a smaller instrument available from Bruker (IRCube) with the same OPD as the EM27 but with a higher level of integration of the components and a smaller throughput (beam diameter). We have chosen for the more spacious EM27, since we needed to replace optics and electronics parts without major mechanical integration issues.

By significantly reducing the spectral resolution, we would lose the ability to resolve individual absorption lines. Doing so, we would no longer leverage the contrast between absorption line and continuum and run the risk to confound spectral structures of surface albedo with atmospheric signal. Wilzewski et al. 2020, showed that spectrally degrading spectra of the GOSAT satellite leads to worse $XCO_2$ consistency with ground-truth measured by TCCON, but also to biases correlating with surface albedo and particle scattering parameters.

Thus, we would not recommend relaxing spectral resolution further for mapping out rather small GHG gradients as showcased here. Worse resolution is acceptable, if the goal is to quantify emissions of localized hotspots with large GHG enhancements, as illustrated by our study using a hyperspectral camera (Knapp et al., 2023).

**(3) Polarization sensitivity?**

**EM27/SUN for the direct sun does not care the input light polarization. However, surface reflected light and aerosol scattered light are polarized. Aerosol scattering is a major contamination source for slant viewing measurement over megacities. Do the authors characterize the instrument polarization? Do they try to measure the polarization of the input light by installing and rotating the polarizer in the front optics?**

FTS are in principle sensitive to the polarization of incoming light, however the effect is less severe compared to grating spectrometers. The main effect for FTS is the polarization dependent reflectivity of the beamsplitter (e.g. (Griffiths and de Haseth, 2007) chapter 5.7). Following the Fresnel equations, the reflectivity of the beamsplitter differs for light polarized parallel and perpendicular to the plane of reflection. In our case this reduces the throughput for light polarized in the plane of the interferometer (horizontally). In laboratory measurements we found that the EM27/SCA is roughly 10% less sensitive to horizontally polarized light compared to vertical polarization (see Fig. AC2), in the spectral range 5600-8000 $cm^{-1}$, relevant for the $CO_2$, $CH_4$ and $O_2$ retrieval windows (W1-W5). The sensitivity difference increases towards lower wavenumbers. Considering that aerosol and surface reflection only polarize the light partially, we consider this a minor issue.

[Figure]

*Figure AC2: Polarization sensitivity of the EM27/SCA. We show the ratio of averaged spectra recorded with horizontally and vertically polarized light.*

Since, at this point, our retrieval is a transmittance calculation neglecting any scattering effects, we do not expect that polarization of the scattered light has an impact on our retrievals. Or in other words: the entire neglect of scattering causes much larger errors than the polarization effects through scattering. In first order, variable polarization of the incoming lightbeam would only lead to broadband transmittance changes, which are absorbed in the background polynomial.

We include the polarization characterization in section 2 and note its possible relevance when retrieving aerosol properties in the future (discussion section of the revised manuscript).

**SPECIFIC COMMENTS**

**(1) P11, Line 189 "geometric assumptions, … not uniform"**

**It is not clear. More detailed description is needed.**

Thank you for the feedback. We updated L189f. of the manuscript with a more explicit explanation.

**(2) P14, Lines 283-284, "SNR …. is most compact"**

**How the authors mean by "compact"? Does it mean calculated SNR has low variation?**

Yes, compact means that SNR has low variation around the expected SZA dependence. Expected variation for reflector measurements is a sqrt(cos(SZA)) relationship, because the signal depends on the angle under which the Lambertian reflector plate is illuminated.

**(3) Page 19 Figure 14 and Page 21, Figure 15,**

**Discussion on wind speed and direction, possible CO2 and CH4 emission sources and ground measured surface pressure in the LA basin will improve the readers understanding. Does the TCCON data at Caltech, Pasadena show the similar trend of the diurnal variation of that day with the West Pasadena data?**

Ad Figure 14: As discussed in lines 302 ff., we cannot interpret the diurnal variation shown in Figure 14, as the main effect introducing variablility is light scattering by aerosols.

Ad Figure 15: For the CH$_4$/CO$_2$ ratio presented in Figure 15, this is different: The TCCON station at Caltech is approximately 5 km away from the WP target location. The afternoon enhancement of CH$_4$/CO$_2$ we see in the WP measurements is also visible in the TCCON data (see Fig. AC3). However, a quantitative comparison between the partial VCD measured in reflected-sun geometry and the total column measurement of TCCON is not straightforward. The amplitude of the CH$_4$/CO$_2$ signal differs substantially between the measurement geometries. So, despite the high precision of the TCCON measurements, the signal is barely larger than the data scatter. While, for the less precise EM27/SCA data, the signal is clearly observable. This illustrates the value of the reflected sunlight measurement geometry.

Regarding information on wind speed and direction, these provide useful information, indeed, and the comment pointed us to an inconsistent interpretation in the manuscript. Looking at the NOAA High-Resolution Rapid Refresh (HRRR) dataset, we find mostly wind directions coming from the south-west / west direction, where major CH$_4$ sources are located (see Fig. AC4 below).

We adjusted our discussion of Figure 15 accordingly and added Fig. AC3 and AC4 to the appendix of the revised manuscript. Thank you for the input.

[Figure]

*Figure AC3: Qualitative comparison of CH$_4$/CO$_2$ ratio between partial VCD below instrument level (upper panel) measured with EM27/SCA (orange dots) and CLARS-FTS (orange triangles) and total VCD measured by TCCON at CalTech, Pasadena (lower panel, (Wennberg et al., 2022)).*

[Figure]

*Figure AC4: CH$_4$ point sources as registered in the California Air Resources Board (CARB) inventory (CARB Pollution Mapping Tool v2.6, 2023). Point size corresponds to CH$_4$ emission strength, color corresponds to the CH$_4$/CO$_2$ emission ratio for the respective CH$_4$ source.*

**TECHNICAL CORRECTIONS**

**(1) Page 16, Figure 11 "RMS"**

**Dese it mean "RMS of SCD.**

RMS refers to the root mean square error of the spectral residuals, as introduced in line 248. We will specifically write "RMS of the spectral residuals for measurements between […]" in the caption of Fig. 11.

**(2) Page 24, References. Journal title abbreviation**

**Examples for reference types are available at https://www.atmospheric-measurement-techniques.net/submission.html.**

You are correct, we corrected this in the revised manuscript, thank you.

**References**

CARB Pollution Mapping Tool v2.6: https://www.arb.ca.gov/carbapps/pollution-map, last access: 3 August 2023.

Griffiths, P. R. and de Haseth, J. A.: Fourier Transform Infrared Spectrometry, 2. ed., Wiley-Interscience, 704 pp., https://doi.org/10.1002/047010631X, 2007.

Wennberg, P. O., Roehl, C. M., Wunch, D., Blavier, J.-F., Toon, G. C., Allen, N. T., Treffers, R., and Laughner, J.: TCCON data from caltech (US), release GGG2020.R0, 2022.

---

## Author Comment (AC2)

**RC2: 'Comment on amt-2023-101', David Griffith, 17 Jul 2023**

https://doi.org/10.5194/amt-2023-101-RC2

**The California Laboratory for Atmospheric Remote Sensing (CLARS) – Fourier Transform Spectrometer is an established facility located at Mt Wilson above the Los Angeles Basin. CLARS maps CO2 and CH4 in the urban atmosphere above LA by measurement of solar radiation scattered from targets around the basin below the observing station. Sequential measurements from a range of targets provides a mapping of CO2 and CH4 concentrations across the basin which can be used through meteorological modelling to elucidate local sources and sinks.**

**CLARS is a large, fixed spectrometer and thus not mobile or usable at other sites. This paper describes a valuable extension of the measurement principle to using a portable, low resolution FTS (modified Bruker EM27) called the EM27/SCA that can in principle be deployed at other suitable sites with little infrastructure requirement. The paper provides a good technical description of the EM27/SCA, an assessment of SNR and measurement precision, and a month of side-by-side measurements beside CLARS in the LA basin to assess overall performance.**

**This presentation is of high quality, clear and well presented, and certainly well suited to publication in AMT. However I feel it stops a bit short of its eventual purpose and usefulness and would be a much more valuable paper if the analysis could be extended as described in the general comments below. I have also made specific and minor technical comments directly in the attached pdf version of the manuscript.**

We thank the reviewer for the appreciation of our work and the helpful comments. Find our point-by-point reply below.

**General comments**

**1. The neglect of scattering in the RemoTeC retrieval of vertical column densities (VCDs) from spectra leads to significant artificial variability in retrieved VCDs as seen clearly in Fig 14. These are large enough to mask the true variability which is the ultimate aim of these measurements. This is most clearly seen in O2, which should be constant. Why has the analysis stopped at this point, with the neglect of scattering, when it is the main identified cause of the inaccuracy in the measurements? To my knowledge RemoTeC can handle scattering in its forward model, so why not include it? This may not be feasible, in this case the authors should explain why. If it is feasible, Figures such as Fig 14 would be immensely more informative because at present most of the diurnal variability we see is artefact; geophysical features of interest are masked by the artefact.**

The retrievals do not include scattering treatment because the radiative transfer model in RemoTeC cannot treat observers situated inside the atmosphere. It can only handle satellite observers. This is due to the fact that the derivatives that populate the Jacobian matrix are calculated via forward adjoint perturbation theory (Hasekamp and Landgraf, 2002, 2005; Hasekamp and Butz, 2008), which is computationally efficient but only little flexible. Updating the code either by implementing observer positions inside the atmosphere or exchanging the entire radiative transfer model is a major conceptual and programming effort. But, we are currently working on an approximate implementation to overcome this limitation which, however, is not operational yet and beyond the scope of our instrument-focused publication here.

Most of the CLARS-FTS studies have used non-scattering retrievals (Fu et al., 2014; Wong et al., 2015, 2016; He et al., 2019; Zeng et al., 2023), and in particular the $CH_4/CO_2$ ratio. $CH_4$ and $CO_2$ undergo approximatly the same lightpath modifications by aerosol scattering. Thus, the scattering effects

tend to cancel in the ratio of both. In that context, Fig. 15 shows actual GHG variability and is, methodologically, in line with previous CLARS-FTS studies.

Thus, we believe that our current analyses are valuable and do illustrate the potential of the EM27/SCA to be a portable companion of the CLARS-FTS. But the manuscript clearly highlights the need for developing retrieval techniques that correct for the effects of scattering. The revision includes a more detailed discussion on the need for the treatment of scattering.

**2. Although the principal aim of the paper is technical, it would benefit at the end with some simple interpretation of the observed variability (after allowing for the scattering arefacts in 1 above) in terms of local sources and sinks using wind and other meteorological data. This is the ultimate aim of the work, but underrepresented in the present version. A full analysis, say with tomography, is outside the scope of the paper, but some simplified interpretation would improve the paper considerably.**

The revision elaborates further on the case study illustrated in Fig. 15 where we find enhanced $CH_4/CO_2$ that is most likely attributable to upwind $CH_4$ sources. A quantitative assessment of source/sink patterns would require longer periods of observation and atmospheric modelling to disentangle sources from sinks and both from transport related effects. Our month-long demonstration deployment aims at demonstrating the working principle and the basic performance parameters of the EM27/SCA measurement technique and at identifying further developments needed e.g. in radiative transfer modelling. Next, we will refine the instrument and the retrievals based on the lessons learned during the demonstration deployment and then aim for a permanent deployment of the EM27/SCA.

**3. CO precision. The low CO precision is ascribed to low SNR due to the low throughput near the detector cutoff wavelength. What is the impact of solar CO on the retrievals and their accuracy and precision? Much of the solar spectrum is dominated by CO lines at very different conditions (temperature, pressure), and the solar CO lines from different parts of the solar disk are also shifted in wavelength relative to terrestrial CO lines. What is the impact of solar CO on the terrestrial CO retrieval?**

We use the full-disk Fraunhofer spectrum assembled by (Toon, 2015) (also recently used by (Coddington et al., 2021) for the TSIS-1 HSRS), that includes the solar CO lines. The assumption is that the solar CO lines are well represented and we have no indications from our analysis here and in previous studies, e.g. using TROPOMI spectra, that solar CO line positions or line shapes are erroneous. Even if there are uncertainties, the reviewer makes the point that the solar CO lines are spectrally shifted with respect to the telluric CO lines and that solar lines are much broader due to the higher temperatures. So, there is no direct spectral correlation between a solar CO line error and the fit to the telluric CO lines. But, of course, there could be indirect correlations via the interfering telluric $CH_4$ and $H_2O$ lines. All these effects are, however, of systematic nature, while Fig. 12 and 13 clearly illustrate that noise is the limiting factor for CO. Thus, at the current level of precision for CO, we expect no impact of potentially erroneous solar CO line properties on our telluric CO retrieval.

**Regarding RC2 specific and minor technical comments (supplement PDF comments)**

We adopted your wording corrections, thank you for that. Please find our response to your other inline comments below.

**L84: Do you mean adapted from the EM27 camtracker, ie you still use the camtracker mirrors but with manual pointing rather than solar tracking?**

Yes, we use a similar alt-azimuthal rotating mirror setup with manual pointing.

**L208: Does NCEP really provide O2 profiles? O2 is effectively constant for these purposes, and it is not listed in the NCEP 2000 reference.**

We derive our $O_2$ a priori profile from the pressure information in the NCEP dataset.

**L253: Could this simply be the (in)accuracy of the O2 line parameters? Was the O2 continuum included in the fit?**

We do not think this is the issue, as retrievals of CLARS-FTS spectra convolved with the EM27/SCA ILS do not show increased residuals in W1. Besides the 5th order polynomial accounting for the spectral continuum, we fit a pseudo-absorber with corresponding cross sections to account for $O_2$ collision induced absorption (c.f. Table 1). We added a sentence in the revised manuscript to clarify this.

**Fig. 12: Could you add a horizontal zero line to each plot? This would make it immediately easier for the reader to assess the differences CLARS-EM27.**

This is indeed helpful, we added the horizontal zero line to each plot. Thank you for the input.

We want to emphasize though, that this figure does not contain data from CLARS-FTS. As described in the caption and L265f. (Sect. 5.2), the differences shown are between EM27/SCA data and its own 30min rolling average to assess the scatter.

**L286: Do you see similar differences to the CLARS data retrieved with the CLARS retrieval code? In general it would be informative to add the CLARS retrievals to this comparison. In Fig 13, I assume the CLARS spectra are retrieved with RemoTeC, in which case these are essentially a replot of the data of Fig 12.**

Indeed Fig. 13 uses CLARS spectra retrieved with RemoTeC. We looked additionally at the correlation between EM27/SCA RemoTeC retrievals and CLARS-FTS retrievals using their standard retrieval algorithm (modified GFIT, see (Fu et al., 2014)). Fig. AC5 shows the correlation plots for $CO_2$ and $CH_4$. For $O_2$ however the target measurements show a substantial bias. It is unclear why this bias occurs. We speculate that this difference could be a result of our choice of regularization. With total column scaling, errors due to the neglect of scattering would also be attributed to regions with lower pressure where absorption lines saturate faster, leading to an underestimation of the total SCD. However, we currently have no proof for this hypothesis. When switching to the lower-partial-column retrieval as described in the manuscript, we found a decrease in the total SCD for target measurements. This effect was strongest for $O_2$.

[Figure]

*Figure AC5: Correlation of $O_2$, $CO_2$, $CH_4$ and CO SCDs (top to bottom) between simultaneous EM27/SCA and CLARS-FTS measurements. EM27/SCA SCDs are retrieved with RemoTeC, while CLARS-FTS SCDs are retrieved with both RemoTeC (bright dots) and the CLARS-FTS standard retrieval software (modified GFIT (Fu et al., 2014), dark dots). This figure was produced in the same way as Fig. 13 in the manuscript.*

**L296: Why is CLARS precision higher? Lower resolution generally leads to higher precision (lower noise bandwidth), so what else contributes? Throughput?**

This is correct. Due to the higher throughput of CLARS-FTS and longer co-adding the spectral SNR of both instruments is comparable. CLARS-FTS has a larger beam diameter at similar FOV. Additionally CLARS-FTS measurements integrate over 3 min, while we report 1 min measurements. The higher spectral resolution leads to deeper absorption features. This in turn results in more precise retrievals at similar SNR.

**References**

Coddington, O. M., Richard, E. C., Harber, D., Pilewskie, P., Woods, T. N., Chance, K., Liu, X., and Sun, K.: The TSIS-1 Hybrid Solar Reference Spectrum, Geophys. Res. Lett., 48, e2020GL091709, https://doi.org/10.1029/2020GL091709, 2021.

Fu, D., Pongetti, T. J., Blavier, J. F. L., Crawford, T. J., Manatt, K. S., Toon, G. C., Wong, K. W., and Sander, S. P.: Near-infrared remote sensing of Los Angeles trace gas distributions from a mountaintop site, Atmos. Meas. Tech., 7, 713–729, https://doi.org/10.5194/amt-7-713-2014, 2014.

Hasekamp, O. P. and Butz, A.: Efficient calculation of intensity and polarization spectra in vertically inhomogeneous scattering and absorbing atmospheres, Journal of Geophysical Research: Atmospheres, 113, https://doi.org/10.1029/2008JD010379, 2008.

Hasekamp, O. P. and Landgraf, J.: A linearized vector radiative transfer model for atmospheric trace gas retrieval, J. Quant. Spectrosc. Ra., 75, 221–238, https://doi.org/10.1016/S0022-4073(01)00247-3, 2002.

Hasekamp, O. P. and Landgraf, J.: Linearization of vector radiative transfer with respect to aerosol properties and its use in satellite remote sensing, J. Geophys. Res. Atmos., 110, https://doi.org/10.1029/2004JD005260, 2005.

He, L., Zeng, Z.-C., Pongetti, T. J., Wong, C., Liang, J., Gurney, K. R., Newman, S., Yadav, V., Verhulst, K., Miller, C. E., Duren, R., Frankenberg, C., Wennberg, P. O., Shia, R.-L., Yung, Y. L., and Sander, S. P.: Atmospheric Methane Emissions Correlate With Natural Gas Consumption From Residential and Commercial Sectors in Los Angeles, Geophys. Res. Lett., 46, 8563–8571, https://doi.org/10.1029/2019GL083400, 2019.

Toon, G. C.: Solar Line List for the TCCON 2014 Data Release, , https://doi.org/10.14291/TCCON.GGG2014.SOLAR.R0/1221658, 2015.

Wong, C. K., Pongetti, T. J., Oda, T., Rao, P., Gurney, K. R., Newman, S., Duren, R. M., Miller, C. E., Yung, Y. L., and Sander, S. P.: Monthly trends of methane emissions in Los Angeles from 2011 to 2015 inferred by CLARS-FTS observations, Atmos. Chem. Phys., 16, 13121–13130, https://doi.org/10.5194/acp-16-13121-2016, 2016.

Wong, K. W., Fu, D., Pongetti, T. J., Newman, S., Kort, E. A., Duren, R., Hsu, Y. K., Miller, C. E., Yung, Y. L., and Sander, S. P.: Mapping CH4 : CO2 ratios in Los Angeles with CLARS-FTS from Mount Wilson, California, Atmos. Chem. Phys., 15, 241–252, https://doi.org/10.5194/acp-15-241-2015, 2015.

Zeng, Z.-C., Pongetti, T., Newman, S., Oda, T., Gurney, K., Palmer, P. I., Yung, Y. L., and Sander, S. P.: Decadal decrease in Los Angeles methane emissions is much smaller than bottom-up estimates, Nat. Commun., 14, 5353, https://doi.org/10.1038/s41467-023-40964-w, 2023.